# Comparative Analysis of Physical Fitness in Aquatic and Terrestrial Environments Among Elderly Women

**DOI:** 10.3390/ijerph22010033

**Published:** 2024-12-30

**Authors:** Frano Giakoni-Ramírez, Josivaldo de Souza-Lima, Catalina Muñoz-Strale, Nicolás Hasche-Zunino, Constanza Sepúlveda-Barría, Andrés Godoy-Cumillaf

**Affiliations:** 1Facultad de Educación y Ciencias Sociales, Instituto del Deporte y Bienestar, Universidad Andres Bello, Las Condes 7550000, Chile; frano.giakoni@unab.cl (F.G.-R.); josivaldo.desouza@unab.cl (J.d.S.-L.); catalina.munoz@unab.cl (C.M.-S.); n.haschezunino@uandresbello.edu (N.H.-Z.); c.seplvedabarra2@uandresbello.edu (C.S.-B.); 2Grupo de Investigación en Educación Física, Salud y Calidad de Vida (EFISAL), Facultad de Educación, Universidad Autónoma de Chile, Temuco 4780000, Chile

**Keywords:** aquatic exercise, land-based exercise, flexibility, strength, older adults, functional fitness, aging

## Abstract

(1) Background: Aging is associated with a progressive decline in physical capacity, which is further exacerbated by conditions such as arthritis and chronic joint pain. This study aimed to compare the effects of aquatic and land-based exercise on the functional fitness of older adult women. (2) Methods: Sixty older women (mean age 66.9 ± 3.8 years) participated in this study, divided into two groups: aquatic exercise and land-based exercise. Both groups completed functional fitness tests, including flexibility (Back Scratch and Chair Sit-and-Reach Tests), lower body strength (Chair Stand Test), and dynamic balance (8-Foot Up-and-Go Test). Statistical analyses compared group performance. (3) Results: Improvements in flexibility were observed in the aquatic group, with trends toward significance for the Back Scratch Test (−0.2 ± 1.0 cm vs. −2.0 ± 0.0 cm, *p* = 0.08) and the Chair Sit-and-Reach Test (2.87 ± 2.0 cm vs. 0.27 ± 1.0 cm, *p* = 0.07). No statistically significant differences were observed between the groups in measures of lower body strength (Chair Stand Test: 19.1 ± 4.47 reps vs. 18.97 ± 3.77 reps, *p* = 0.9) or dynamic balance (8-Foot Up-and-Go Test: 6.28 ± 6.2 s vs. 6.03 ± 5.83 s, *p* = 0.07). (4) Conclusions: Aquatic exercise showed greater improvements in flexibility, particularly in the upper and lower body, although these differences did not reach statistical significance. Both training modalities were equally effective in maintaining lower body strength and dynamic balance in older adult women. These findings support the inclusion of tailored exercise programs in aging populations to address specific functional needs.

## 1. Introduction

Aging is a natural and multifaceted process that involves a series of biological, psychological, and social changes [1]. As people age, there is a progressive decline in functional capacity, particularly in the musculoskeletal and cardiovascular systems, significantly affecting mobility, strength, and physical endurance [2]. These changes considerably impact the quality of life and independence of older adults, increasing the risk of falls, chronic diseases, and functional dependency. In this context, it becomes essential to explore and promote physical activities tailored to the conditions of this population, aiming to improve or maintain their physical capacity in a safe and effective manner [3,4]. Both aquatic exercise (AE) and land-based exercise (LBE) are considered beneficial interventions to encourage physical activity in older adults. However, the differences in the impact of these two environments on physical capacity require further investigation to fully understand their effects and optimize rehabilitation and physical maintenance programs for this population [5,6].

The progressive loss of functional capacity in older adults is not only an inevitable consequence of aging but is also closely linked to the increase in chronic conditions such as sarcopenia, osteoarthritis, and cardiovascular deterioration [7,8,9]. These conditions limit mobility and reduce strength and endurance, affecting the ability to perform daily activities independently [10,11]. Among the common complications in this stage of life are decreased flexibility, balance, and coordination—factors that increase the risk of falls and injuries, one of the most serious public health issues for this age group [12,13]. In this regard, both AE and LBE have been proposed as suitable interventions to mitigate these effects. However, there is a need for in-depth research on how each of these environments impacts the various components of physical capacity to optimize exercise programs tailored to the specific needs of older adults [6,14,15].

AE and LBE offer unique benefits for improving the physical capacity of older adults, although each has specific characteristics that can influence outcomes [6,14,16]. LBE, for instance, often includes low-impact activities such as walking and resistance exercises that enhance muscular strength and stability [17,18,19]. However, the weight-bearing load on joints can be limiting for those with conditions such as arthritis or joint pain [20,21,22]. In contrast, the aquatic environment reduces joint impact due to buoyancy, facilitating movement for individuals experiencing pain or mobility restrictions [23,24,25]. This setting also enables cardiovascular endurance training in a safe manner, which is essential for maintaining functional capacity in older adults [26]. Nevertheless, aquatic exercise can present challenges in breath control and temperature regulation, factors that must be considered when designing training programs [27,28,29].

The early identification and management of functional capacity issues in older adults are essential to prevent complications such as the loss of independence, increased fall risk, and the progression of chronic diseases [30]. Despite evidence supporting the benefits of physical exercise in maintaining and improving functional health, a gap remains in the implementation of programs tailored to the specific needs of this population [31]. Designing strategies that account for the limitations and strengths of older adults is crucial to ensuring the safety and effectiveness of interventions. In this context, understanding the differences in the effects of land-based and aquatic exercise could provide valuable insights for optimizing training and rehabilitation programs, promoting active and healthy aging.

While physical exercise in land-based and aquatic environments has been widely recommended for older adults, there remains a lack of comparative evidence specifically evaluating how each of these settings impacts different components of physical capacity, such as strength, endurance, flexibility, and balance. This gap in knowledge limits the development of personalized strategies to maximize the benefits of exercise for this population. Therefore, the present study aims to compare the physical capacity of older adult women in land-based and aquatic environments, using a quantitative approach to explore the differences and similarities in the outcomes achieved in both settings.

## 2. Materials and Methods

### 2.1. Study Design

This descriptive study was conducted with the approval of the Scientific Ethics Committee of the Universidad Autónoma de Chile (CEC 2320). All participants were informed about the purpose and procedures of the study and provided written informed consent to participate. The study protocol adhered to the ethical principles outlined in the Declaration of Helsinki.

### 2.2. Study Participants

The participants of this study were recruited from the senior program of a city in Chile, in collaboration with recreational workshops offered by local senior recreation centers. These workshops provide activities tailored to women aged 60 years or older, conducted in both aquatic and land-based settings. The research team identified and personally contacted the centers, obtaining the necessary authorizations from workshop coordinators. The flowchart in Figure 1 details participant eligibility, group assignment, and overall study design.

The inclusion criteria required participants to be self-sufficient women aged 60 years or older, with at least 60% attendance at the workshops and a minimum of 12 months of participation in these programs. Participants were excluded if they failed to attend the day of testing, did not adhere to the exercise protocol, experienced physical discomfort, had disabling illnesses, or presented permanent or temporary contraindications for physical activity.

This study focused exclusively on female participants to account for sex-specific differences in physiological responses to exercise, particularly in older adults. Women tend to experience distinct patterns of muscle strength loss, flexibility decline, and balance changes compared to men, which may influence the outcomes of exercise interventions. This homogeneous sample ensures the results are specific and relevant to designing exercise programs for older women.

#### Additional Information

No additional physical activities outside the study were reported by participants during the evaluation period.

Chronic health conditions or pre-existing conditions were not controlled in this study, though participants were required to be free of contraindications for physical activity.

The study participants were divided into two groups based on the training modality performed. Group 1 consisted of older adults engaged in functional exercise conducted indoors in recreational centers. These exercises focused on full-body movements designed to improve strength, flexibility, and stability. Each session lasted 50–60 min and was conducted 2–3 times per week. Group 2 included older adults participating in aquatic activities like conventional hydro-gymnastics. These activities were recreational and adapted, emphasizing low-impact movements suitable for older adults; the aquatic sessions lasted 50 min and were performed 2–3 times per week. It is important to note that the aquatic activities did not involve specific swimming techniques or open-water exercises. Both groups perceived the intensity of the sessions as moderate, as assessed using the Borg Rating of Perceived Exertion Scale. The Borg RPE Scale is a widely validated and reliable tool for quantifying subjective effort during physical activities, particularly in populations such as older adults where objective measures of intensity may not always be feasible [32].

### 2.3. Procedure

The evaluation was conducted at the participating recreational centers with the support of an external evaluator, who performed the measurements without prior knowledge of the participants’ intervention groups. Descriptive data were collected, including age, weight, height, and program participation duration (minimum of 12 months). Each participant also completed a physical fitness assessment using the Senior Fitness Test protocol [33]. During the collection of sociodemographic data, participants were asked whether their activities were conducted in an aquatic or land-based setting. Based on this information, two groups were formed for comparison: the AE and the LBE. The primary outcome measures included assessments of lower and upper body strength, aerobic endurance, flexibility, agility, and dynamic balance.

The selection of a 2 kg weight for the arm strength test (Arm Curl Test) aligns with established protocols validated in studies involving older adult women. This weight is considered optimal for assessing upper body strength in this population, as it provides sufficient resistance to measure muscular endurance without causing undue strain or fatigue, which is critical for ensuring participant safety during evaluations [33]. Similarly, the use of a 30 s time interval for strength tests, such as the Chair Stand Test and the Arm Curl Test, is grounded in its widespread adoption in gerontological fitness research. This interval effectively balances the need to capture a reliable measure of functional capacity while minimizing the risk of overexertion in older adults [33]. These standardized parameters enhance the comparability of findings across studies and ensure methodological consistency in evaluating physical fitness components in aging populations.

Evaluation Methodology


**Outcome**

**Test**

**Procedure**

**Measurement**
Leg StrengthChair Stand TestParticipants began seated on a chair without armrests, feet flat on the ground, hands crossed over the chest. They stood up and sat down as many times as possible within 30 s.Total number of repetitions completed.Arm StrengthArm Curl TestParticipants held a 2 kg dumbbell in one hand, arm extended along the body. They performed elbow flexions, bringing the dumbbell to the shoulder, as many times as possible within 30 s.Total number of repetitions completed.Aerobic Endurance2 Min Step TestParticipants stood next to a wall where a height marker (half the distance between hip and knee) was placed. They alternately raised their knees to reach the marked height for two minutes.Total number of knee lifts meeting the height criterion.Leg FlexibilityChair Sit-and-Reach TestSeated at the edge of a chair, participants extended one leg forward (heel on the ground, toes pointing upward) and attempted to reach or surpass the tip of the toes using both hands.Distance in cm (positive if toes reached/exceeded, negative if not).Shoulder FlexibilityBack Scratch TestWhile standing, participants placed one hand over the shoulder and down the back while the other hand was placed behind the back and extended upward, attempting to touch the fingertips of both hands.Distance in cm (positive if fingers overlapped, negative if not).Agility and Dynamic Balance8-Foot Up-and-Go TestParticipants began seated on a chair. At the evaluator’s signal, they stood up, walked around a cone placed 2.44 m away, and returned to the chair to sit down.Time in seconds to complete the course.

### 2.4. Statistical Analysis

Descriptive statistics were expressed as mean (standard deviation) and median (interquartile range) for continuous variables. Physical assessment results and descriptive data were compared using *t*-tests for means and Mann–Whitney U tests for medians, depending on the data distribution. Prior to the statistical analyses, normality tests, including the Kolmogorov–Smirnov test, were applied to assess the distribution of continuous data. All analyses were performed using JAMOVI software, version 2.3.18 (The Jamovi Project, 2022). The significance level was set at *p* < 0.05.

## 3. Results

### 3.1. Descriptive Characteristics of the Participants

A total of 60 female older adults were included in this study, equally divided between two groups: 30 participants in the LBE group and 30 in the AE group. Table 1 presents a comprehensive overview of the participants’ baseline physical characteristics and functional test outcomes.

Participants in the LBE group had slightly higher mean weight (68.33 ± 7.27 kg) compared to the AE group (66.93 ± 6.33 kg), though both groups showed similar variability across the minimum and maximum values. Height was marginally greater in the AE group (162.10 ± 6.16 cm) compared to the LBE group (159.10 ± 5.14 cm). Both groups displayed comparable mean ages, with minimal differences (LBE: 67.03 ± 4.10 years; AE: 66.83 ± 3.37 years).

For functional test outcomes, the AE group showed a slightly higher median performance in the Chair Sit-and-Reach Test (2.87 cm vs. 0.27 cm), suggesting better lower body flexibility. However, the LBE group performed marginally better in the Back Scratch Test, with a median of 0.00 cm compared to −1.00 cm in the AE group. These findings align with the expected benefits of land-based exercises for maintaining upper-body flexibility. Similar trends were observed in agility and dynamic balance, with both groups showing comparable performance in the 8-Foot Up-and-Go Test (LBE: 6.03 ± 0.83 s; AE: 6.28 ± 0.83 s), indicating equivalent levels of functional mobility.

### 3.2. Normality Assessment of the Participants’ Data

Table 2 presents the results of the Shapiro–Wilk test used to evaluate the normality of physical and functional test data for both exercise groups. Out of the nine variables evaluated, three variables (weight, height, and age) were normally distributed (*p* > 0.05), while the remaining six variables, including flexibility, agility, and some strength measures, did not meet the criteria for normality (*p* ≤ 0.05).

Table 2 demonstrates the variability in data distribution across the different components of physical fitness assessed. For instance, measures such as the Back Scratch Test and the 8-Foot Up-and-Go Test, which involve dynamic or complex movements, exhibited non-normal distributions, potentially reflecting individual variability in functional abilities.

This analysis allowed for the identification of distribution patterns that guided the selection of statistical tests, applying parametric methods for variables with normal distribution and non-parametric methods for those that did not meet normality criteria. In this way, the analysis was aligned with the characteristics of the data, strengthening the comparisons made between the two groups.

### 3.3. Comparison of Parametric Variables

Table 3 provides the comparison of parametric variables, including age, weight, and leg strength (assessed through the Chair Stand Test), between the AE and LBE groups. These variables were analyzed using *t*-tests to identify potential differences in functional capacity.

The results indicate no statistically significant differences between the groups in terms of age, weight, or leg strength (as measured by the Chair Stand Test). This suggests that the baseline characteristics and functional abilities related to these variables are comparable between the two training environments.

### 3.4. Comparison of Non-Parametric Variables

This study presents the results of the Mann–Whitney U test conducted to evaluate non-parametric variables between the AE and LBE groups. The *p*-values indicate no statistically significant differences across the analyzed variables (Table 4), suggesting comparable outcomes for non-parametric measures between the two exercise environments.

These findings suggest that, while both training environments promote similar outcomes in flexibility, aerobic endurance, and dynamic balance, further research with larger sample sizes is warranted to explore potential trends, particularly for variables like height and the Chair Sit-and-Reach Test.

### 3.5. Graphics: Visual Representation

The delta percentage analysis highlights the relative differences in performance outcomes between the AE and LBE groups (Figure 2). Notable findings include a substantial increase in the Chair Sit-and-Reach Test (975%) for the AE group, indicative of improved flexibility in this environment. Conversely, a significant negative delta was observed in the Back Scratch Test (−90%), suggesting decreased performance in upper body flexibility for the AE group compared to land exercise. Other variables, such as the 2 Min Step Test and age, showed minimal differences, with deltas close to zero, suggesting comparable performance. This visualization effectively showcases how specific training modalities influence physical and functional test outcomes differently.

## 4. Discussion

The present study compared the physical characteristics and functional test outcomes of older adults undergoing AE and LBE. The main results indicate that no statistically significant differences were observed in parametric variables such as weight, age, and performance on the Chair Stand Test (30 s). However, some non-parametric variables, such as height and the Back Scratch Test, showed trends suggesting functional differences associated with the type of training received.

The results related to the Back Scratch Test suggest better flexibility in the AE group, aligning with previous research highlighting the benefits of aquatic environments in reducing joint load and improving functional mobility [14]. Additionally, studies have demonstrated that aquatic exercise significantly enhances upper body flexibility compared to LBE, as observed in a thermal AE protocol that resulted in a 25.8% improvement in the Back Scratch Test [34]. Conversely, the similar outcomes between groups in the Chair Stand Test are consistent with studies indicating that both environments are effective in maintaining lower body strength, as water and land-based exercise can preserve knee extension strength in older adults [35]. However, research also emphasizes that aquatic exercise may be more beneficial for improving dynamic balance—a critical factor for fall prevention in older adults—while LBE may provide advantages in overall muscular endurance [14,34].

The biomechanical and physiological characteristics of aquatic exercise offer measurable and distinct benefits compared to land-based exercise. In water, buoyancy reduces joint forces by up to 90%, facilitating dynamic exercises with lower injury risk and greater safety for older adults with osteoarthritis [36]. For instance, aquatic protocols have shown increases of 25.8% in upper body flexibility tests such as the Back Scratch Test, whereas changes with LBE are limited to a range of 12–15% [34]. Additionally, hydrostatic pressure improves venous return, leading to significant reductions in systolic blood pressure by approximately 5 mmHg after moderate-intensity aquatic exercise sessions [37]. On the other hand, LBE provides specific benefits in terms of strength and postural stability. For example, land-based programs have demonstrated up to a 20% improvement in knee extension strength, a critical ability for functional activities such as rising from a chair [38]. These quantitative differences highlight the importance of selecting the most suitable environment based on individual needs and therapeutic goals.

The choice of training environment, whether aquatic or land-based, has unique implications for designing exercise programs for older adults. Aquatic programs have proven particularly effective in enhancing functional abilities such as balance and postural mobility, due to the safety and support provided by water, which reduces fall risk and increases participants’ confidence in their movements [39]. Furthermore, a study found that water exercises with floating devices promote social interaction and improve self-efficacy in older adults with arthritis [40]. Conversely, LBE has been associated with specific benefits in muscle strength and endurance, with significant improvements in tests such as the “Two-Minute Step Test” and the “30 s Chair Stand” [41]. Integrating both types of training into a combined program can provide a comprehensive strategy to optimize both physical functionality and psychological well-being in older adults [42].

This study has certain limitations that should be considered when interpreting the findings. The small sample size may limit the generalizability of the results to other populations of older adults with varying levels of functionality or health conditions. Additionally, external variables such as physical activity performed outside the evaluated programs were not controlled, which may have influenced the observed outcomes. Individual factors, such as pre-existing conditions and adherence to interventions, may also have affected the programs’ effectiveness. Furthermore, the lack of a strict control group limits the ability to directly attribute the observed changes to the interventions. This highlights the need for more robust experimental designs, such as randomized controlled trials with larger sample sizes. Future research could incorporate multimodal functional assessments, combining objective and self-reported measurements, to provide a more comprehensive analysis of the benefits. Despite these limitations, the findings provide a valuable foundation for designing targeted programs that optimize functional health and the quality of life in older adults.

Future research should also focus on longitudinal studies to provide a more detailed evaluation of the sustained effects of aquatic and land-based exercise on different dimensions of functional capacity. Integrating advanced techniques, such as biomechanical analysis and high-precision physiological measurements, could shed light on the specific mechanisms underlying improvements in flexibility, strength, and other motor capacities observed in these interventions. Additionally, it would be valuable to explore how these strategies impact not only physical performance but also indicators related to quality of life and fall prevention over time.

## 5. Conclusions

This study contributes to understanding the potential benefits of AE and LBE in older adults, demonstrating improved flexibility in the aquatic environment and the comparable effectiveness of both settings in maintaining lower body muscle strength. However, the study’s findings should be interpreted with caution due to limitations, including a small sample size and the lack of control over external variables such as additional physical activities performed outside the training programs. While these results provide preliminary insights, further research is needed to explore the specific needs and characteristics of older adult populations and to evaluate the effectiveness of tailored, evidence-based programs in diverse settings. Such studies would help to optimize interventions aimed at promoting active and healthy aging.

## Figures and Tables

**Figure 1 ijerph-22-00033-f001:**
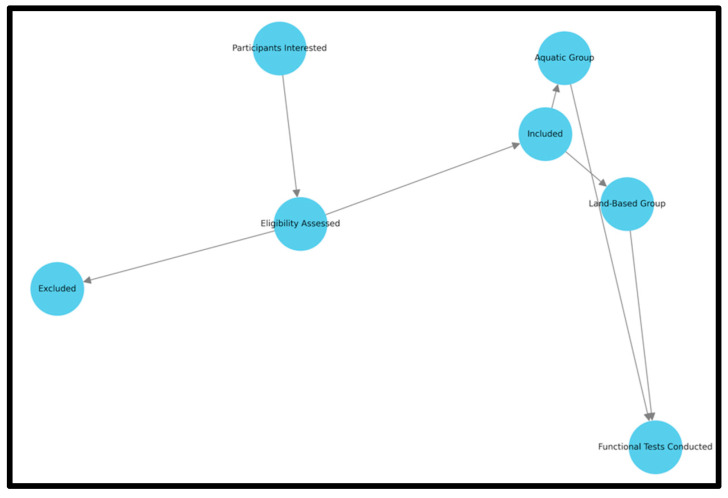
Flowchart of participant eligibility and study design.

**Figure 2 ijerph-22-00033-f002:**
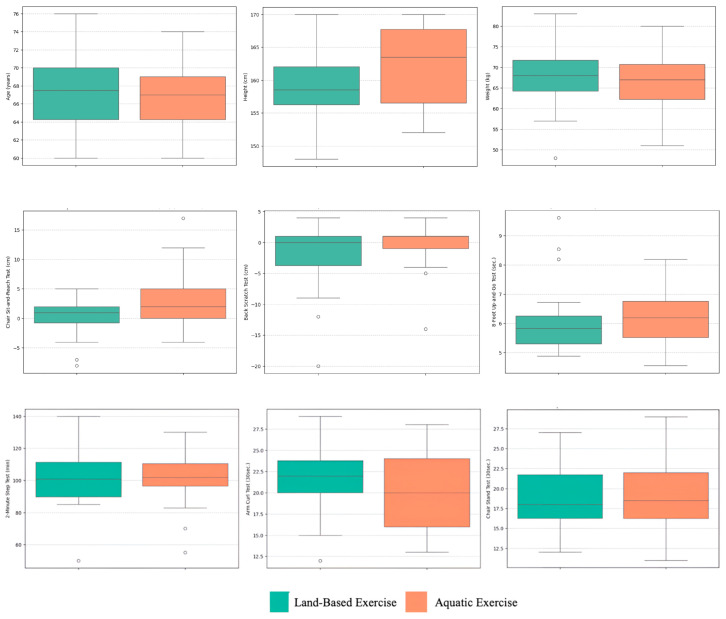
Box plot analysis of functional and physical test variables across training modalities.

**Table 1 ijerph-22-00033-t001:** Physical characteristics and functional test outcomes in older adults by training group.

	G1—LBE *n* = 30				G2—AE Group *n* = 30			
Variable	Mean ± SD	Median	Minimum	Maximum	Mean ± SD	Median	Minimum	Maximum
Weight (kg)	68.33 ± 7.27	68.00	48.00	83.00	66.93 ± 6.33	67.0	51.00	80.0
Height (cm)	159.10 ± 5.14	158.50	148.00	170.00	162.10 ± 6.16	163.5	152.00	170.0
Age (years)	67.03 ± 4.10	67.50	60.00	76.00	66.83 ± 3.37	67.0	60.00	74.0
Chair Stand Test (30 s)	18.97 ± 3.77	18.00	12.00	27.00	19.10 ± 4.47	18.5	11.00	29.0
Arm Curl Test (30 s)	21.57	22.00	12.00	29.00	20.40	20.0	13.00	28.0
2 Min Step Test (min)	101.90 ± 16.96	101.00	50.00	140.00	102.33 ± 15.59	102.0	55.00	130.0
Chair Sit-and-Reach Test (cm)	0.27	1.00	−8.00	5.00	2.87	2.0	−4.00	17.0
Back Scratch Test (cm)	−2.00	0.00	−20.00	4.00	−0.20	1.0	−14.00	4.0
8-Foot Up-and-Go Test (s)	6.03	5.83	4.88	9.62	6.28	6.2	4.56	8.2
SD: standard deviation.								

**Table 2 ijerph-22-00033-t002:** Distribution assessment of physical and functional test data.

Test Name	W Statistic	*p*-Value	Distribution	Type of Test
Weight (kg)	0.972	0.1831	Normal	Parametric
Height (cm)	0.9552	0.0274	Not Normal	Non-Parametric
Age (years)	0.9779	0.3466	Normal	Parametric
Chair Stand Test (30 s)	0.9705	0.1552	Normal	Parametric
Arm Curl Test (30 s)	0.96	0.0469	Not Normal	Non-Parametric
2 Min Step Test (min)	0.951	0.0174	Not Normal	Non-Parametric
Chair Sit-and-Reach Test (cm)	0.9157	0.0005	Not Normal	Non-Parametric
Back Scratch Test (cm)	0.7842	0.000	Not Normal	Non-Parametric
8-Foot Up-and-Go Test (s)	0.9087	0.0003	Not Normal	Non-Parametric

**Table 3 ijerph-22-00033-t003:** Assessment of parametric variables across AE and LBE groups.

Variable	Mean ± SD (AE)	Mean ± SD (LBE)	t-Statistic	*p*-Value
Age (years)	67.03 ± 3.37	67.03 ± 4.10	0.206	0.837
Weight (kg)	66.93 ± 6.33	68.33 ± 7.27	0.796	0.429
Chair Stand Test (30 s)	19.40 ± 4.47	18.97 ± 3.77	−0.125	0.901

**Table 4 ijerph-22-00033-t004:** Assessment of non-parametric variables across AE and LBE groups.

Variable	Median (AE)	Median (LBE)	t-Statistic	*p*-Value
Height (cm)	162.00	159.50	318.50	0.05
Arm Curl Test (30 s)	22.00	21.50	516.50	0.33
2 Min Step Test (min)	102.50	101.50	429.50	0.77
Chair Sit-and-Reach Test (cm)	3.00	2.00	329.00	0.07
Back Scratch Test (cm)	−2.00	−1.00	332.50	0.08
8-Foot Up-and-Go Test (s)	6.20	6.00	325.00	0.07

## Data Availability

The datasets generated and analyzed during the current study are available from the corresponding author on reasonable request.

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
