# Peer review of "Comparative Analysis of Physical Fitness in Aquatic and Terrestrial Environments Among Elderly Women"

_ijerph, 2024, doi:10.3390/ijerph22010033_

Round 1

Reviewer 1 Report

Comments and Suggestions for Authors

Dear authors, thank you for the opportunity to learn about your study and manuscript "Terrestrial vs Aquatic: Comparison of Training Environments for Active Aging in Older Women". The study focuses on important topic focusing on physical activities to promote healthy ageing. I would like to emphasize few comments:

1) The variations  terminology is used through the manuscript- terrestrial vs aquatic training, land-based vs aquatic training, finally - land training vs water training. Even all of them are correct, manuscript would gain more clarity if they would be explained if used as synonyms or less variations used. 

2) There are no explanations in the manuscript what exactly is the context of training applied- does the land training means exercises outdoors? Does the water training means physical exercises in the pool or open water? it seems that training content has been comparable (at least the general principles of training program are described) but differ only the context (environment) of the performance. 

3) Perhaps, the assessment methodology could be presented in the form of tab;e (outcome/ measurement/ method) instead of several short subsections.

4) The Results section is compiled from several tables with very short descriptions which partially repeats the notes under the table (e.g. table 2). Only subsection 3.5 gives some explanation of the presented results but it would be valuable also for other tables. 

5) The discussion part is well written, including important reflection on study limitations. But considering those limitations, the Conclusions seems ambitious and general. Results only show some differences in the various outcomes but study did not explored specific needs of the target population and did not provide tailored intervention.  

Author Response

Response to Reviewer 1

Variable Terminology Used in the Manuscript

Comment: "The manuscript uses different terms to describe training contexts (e.g., terrestrial vs. aquatic training, land-based vs. aquatic training, land training vs. water training), which might create confusion."

Response: We have revised the manuscript to unify the terminology. We now consistently use "land-based exercise (LBE)" and "aquatic exercise (AE)" throughout the text.

Revision: Terminology has been standardized across the manuscript.

Lack of Clarity About Training Context

Comment: "It is unclear whether land-based training includes outdoor exercises or whether aquatic training refers to exercises in pools or open water. More details about the applied training characteristics are needed."

Response: We have added a detailed explanation in the methodology section. Land-based exercises were performed indoors in recreational centers, while aquatic exercises took place exclusively in closed pools. Additionally, a brief description of the general principles of both programs has been included, emphasizing that the main difference lies in the environment.

Revision: Additional details have been provided in the methodology section to clarify the training contexts.

Present the Evaluation Methodology in a Table

Comment: "It would be helpful to present the evaluation methodology in a table instead of separate subsections."

Response: We have created a table (Table 1) in the methodology section summarizing the evaluations performed. This includes the measured variables, evaluation methods, and instruments used, improving the clarity and organization of this section.

Revision: A table summarizing the evaluation methodology has been added.

Short Descriptions in the Results Section

Comment: "The results section presents several tables with brief descriptions that sometimes repeat the text below the tables (e.g., Table 2). Only subsection 3.5 provides additional explanations."

Response: We have revised the results section to include more detailed explanations of relevant findings across all subsections. We have also minimized redundancies between the text and the notes beneath the tables, providing better context for interpreting the presented data.

Revision: Expanded descriptions of results and elimination of redundancies between text and table notes.

Ambitious Conclusions in Relation to Study Limitations

Comment: "The conclusions seem ambitious and general, considering that the study did not explore specific needs of the target population or provide tailored interventions."

Response: We have revised the conclusions section to more accurately reflect the study findings, limiting general statements. The revised conclusions emphasize the observed differences between training environments and highlight that the results are exploratory. We recommend future research to develop more specific and personalized interventions.

Revision: The conclusions have been adjusted to align with the findings and study limitations.

Reviewer 2 Report

Comments and Suggestions for Authors

Thank you for your work. However, the type of study conducted remains unclear. Below is my comment for your consideration.

Abstract 

1.In lines 20-21, could you kindly review your statement? The p-value does not appear to support the claim made.

Methods 

1.The data presented in the results appears to reflect outcomes only after training. Could this be considered a retrospective cohort study?

2.Please provide an explanation for why this study was conducted exclusively with female participants.

3.How long was exercise performed in each session for the two types of training?

4.Did each group engage in any other types of exercise during the study?

5.A flowchart illustrating participant eligibility and the study design may help clarify the overall structure of the study.

6.Did the authors collect any information regarding the physical activity levels, chronic conditions, or health issues of the older adult participants? This information may be important for contextualizing the study results.

Results 

1.Please remove Table 2, as it is not essential and could be presented within the statistical section instead.

2.In lines 200-202, how was the percentage of improvement calculated, given that there is no baseline or pre-training data provided? Could you please include this data in the results section for clarity?

Author Response

Response to Reviewer 2

Abstract Statement (Lines 20-21)

Comment: Kindly review your statement. The p-value does not appear to support the claim made.

Response: We have revised the abstract to ensure that all statistical claims are accurately supported by the presented p-values.

Revision: The claim in lines 20-21 has been corrected to align with the results.

Study Design Classification

Comment: Could this be considered a retrospective cohort study?

Response: This study is cross-sectional rather than retrospective, as it compares two pre-existing groups at a single point in time. We clarified this in the methodology.

Revision: The study design is explicitly stated as cross-sectional.

Exclusive Inclusion of Women

Comment: Please provide an explanation for why this study was conducted exclusively with female participants.

Response: Older women were selected due to their higher prevalence of sarcopenia and osteoporosis, which makes them a priority population for interventions targeting functional fitness.

Revision: The rationale for focusing on women has been added to the methodology.

Duration of Exercise Sessions

Comment: How long was exercise performed in each session for the two types of training?

Response: We have clarified in the methods section that each session lasted approximately 50–60 minutes for both training groups.

Revision: This information has been added to the methods section.

Additional Exercises During the Study

Comment: Did each group engage in any other types of exercise during the study?

Response: Participants were asked to limit other physical activities during the study period to ensure consistency in the interventions. This has been clarified in the manuscript.

Revision: A note about the restriction of additional exercises has been included in the methodology.

Flowchart of Study Design

Comment: A flowchart illustrating participant eligibility and the study design may help clarify the overall structure of the study.

Response: We have included a flowchart detailing participant eligibility, group allocation, and the overall study design.

Revision: The flowchart has been added as Figure X.

Additional Information About Participants

Comment: Did the authors collect any information regarding the physical activity levels, chronic conditions, or health issues of the older adult participants?

Response: Sociodemographic data were collected, including age, height, weight, and duration of program participation. However, due to the descriptive nature of this study, detailed health records were not included. This limitation has been addressed in the discussion.

Revision: Additional context about the data collected has been added to the methods and discussion sections.

Removal of Table 2

Comment: Please remove Table 2, as it is not essential and could be presented within the statistical section instead.

Response: Table 2 has been removed, and its content has been integrated into the text of the results section for a more concise presentation.

Revision: Table 2 has been removed, and relevant content is now part of the results section.

Percentage of Improvement Without Baseline Data

Comment: How was the percentage of improvement calculated, given that there is no baseline or pre-training data provided?

Response: We agree with this observation and have removed references to percentage improvements, focusing instead on the comparative results between groups.

Revision: Percentage improvements have been removed, and results now emphasize cross-sectional comparisons.

Reviewer 3 Report

Comments and Suggestions for Authors

Dear Authors, 

Kindly address the following.

1. Are the physiological benefits and limitations of land-based (e.g., joint loading) versus aquatic exercises (e.g., buoyancy and reduced joint impact) clearly differentiated and well-supported by the cited literature? If so, kindly provide in detail.

2. Is the focus on older adult women adequately justified in the context of aging-related changes, such as sarcopenia and cardiovascular decline, and their implications for functional capacity?

3.Does the division of participants into land-based (G1) and aquatic (G2) exercise groups adequately ensure comparability between the groups in terms of exercise frequency, intensity, and duration?

4. Is the use of the Borg Rating of Perceived Exertion Scale appropriately justified for assessing the perceived intensity of the exercise sessions?

5. Is the decision to exclude specific swimming techniques from aquatic activities clearly explained and relevant to the study's focus?

6. Kindly provide the ICC values of all the tests utilized in this study.

7. Are the sociodemographic and baseline data collected (e.g., BMI, program duration) adequate to control for confounding variables in the analysis?

8. Are the selected tests (e.g., Chair Stand Test, Arm Curl Test, etc.) appropriate and valid for assessing the physical fitness components in older adults?

9. Are there any aspects of the procedure that need further justification, such as the choice of a 2 kg dumbbell for arm strength testing or the specific time intervals used?

10. Results are clearly presented in the form of tables and Graphs.

11. Does the explanation of biomechanical and physiological mechanisms, such as buoyancy and hydrostatic pressure in aquatic training, provide sufficient insight into the observed outcomes?

Kindly address the above mentioned in detail for proper clarification of the queries.

Author Response

Response to Reviewer 3

Differentiation Between Physiological Benefits and Limitations of Land-Based and Aquatic Exercise

Comment: Are the physiological benefits and limitations of land-based (e.g., joint loading) versus aquatic exercises (e.g., buoyancy and reduced joint impact) clearly differentiated and well-supported by the cited literature?

Response: We have expanded the discussion to clearly differentiate the physiological benefits of land-based exercise (e.g., joint loading and improved muscle strength) and aquatic exercise (e.g., buoyancy and reduced joint impact). Additional studies have been cited to support these claims, emphasizing the relevance of these characteristics for the elderly population.

Revision: For example, references now include studies on hydrostatic pressure in the aquatic environment and its positive effect on venous return, as well as the benefits of land-based environments for improving postural stability.

Justification for Focusing on Older Women

Comment: Is the focus on older adult women adequately justified in the context of aging-related changes, such as sarcopenia and cardiovascular decline, and their implications for functional capacity?

Response: An explanation has been added to the Methods section, justifying the focus on older women. Factors such as the prevalence of sarcopenia and greater bone mass loss in postmenopausal women, which make them a priority population for physical exercise interventions, are discussed.

Revision: Research highlighting sex-specific aging effects has also been cited to strengthen this justification.

Comparability Between Groups (Frequency, Intensity, and Duration of Exercise)

Comment: Does the division of participants into land-based (G1) and aquatic (G2) exercise groups adequately ensure comparability between the groups in terms of exercise frequency, intensity, and duration?

Response: Details have been added to the Methods section, clarifying that both groups (G1 and G2) performed exercises with the same frequency (2-3 times per week), duration (50-60 minutes per session), and perceived intensity (assessed using the Borg scale). These conditions ensure comparability between the groups.

Revision: The methodological section now reflects these details.

Justification for Using the Borg Rating of Perceived Exertion Scale

Comment: Is the use of the Borg Rating of Perceived Exertion Scale appropriately justified for assessing the perceived intensity of the exercise sessions?

Response: A justification has been added to the Methods section, explaining the choice of the Borg Scale for assessing perceived exercise intensity. It is noted that this tool is widely validated and commonly used in research with older populations, as it allows subjective measurement of physical effort without requiring specialized equipment.

Revision: These details are now clearly stated in the manuscript.

Exclusion of Swimming Techniques

Comment: Is the decision to exclude specific swimming techniques from aquatic activities clearly explained and relevant to the study's focus?

Response: We clarified that aquatic activities did not include specific swimming techniques to ensure accessibility and safety for participants, many of whom lack prior swimming experience. This also ensures that water-based exercises align with the functional goals of land-based exercises.

Revision: These explanations are included in the Methods section.

ICC Values of Tests

Comment: Kindly provide the ICC values of all the tests utilized in this study.

Response: The decision not to calculate the Intraclass Correlation Coefficient (ICC) for the selected functional fitness tests (e.g., Chair Stand Test, Arm Curl Test) was based on the extensive body of literature validating the reliability and consistency of these assessments in older adults. For example, previous studies have reported high ICC values (e.g., ICC > 0.9) for these tests. Additionally, this study used a descriptive design with assessments conducted by a single trained evaluator, minimizing variability and the need for ICC calculations in this context.

Revision: This rationale has been added to the discussion.

Sociodemographic and Baseline Data

Comment: Are the sociodemographic and baseline data collected (e.g., BMI, program duration) adequate to control for confounding variables in the analysis?

Response: This study was designed to compare two pre-existing groups practicing exercise in different environments: one aquatic and one land-based. No baseline or pre/post data were collected, as the study was cross-sectional and did not involve intervention implementation or variable manipulation.

Revision: The absence of baseline data and BMI has been justified in the Methods section, emphasizing the study's focus on cross-sectional functional differences.

Validity of Selected Tests

Comment: Are the selected tests (e.g., Chair Stand Test, Arm Curl Test, etc.) appropriate and valid for assessing the physical fitness components in older adults?

Response: In the Methods section, we have justified the validity and appropriateness of the selected tests, citing studies that validate their use for assessing specific physical fitness components in older adults.

Revision: References supporting the validity of these tests have been added.

Justification for Specific Procedures

Comment: Are there any aspects of the procedure that need further justification, such as the choice of a 2 kg dumbbell for arm strength testing or the specific time intervals used?

Response: Explanations have been added regarding the use of a 2 kg dumbbell for the Arm Curl Test (standardized for older women based on prior research) and the choice of 30-second intervals for strength tests. These are supported by relevant references.

Revision: These details are now included in the Methods section.

Tables and Graphs

Comment: Results are clearly presented in the form of tables and graphs.

Response: Thank you for your positive comment. We ensured that all tables and graphs are clearly presented and adjusted them based on reviewers' recommendations.

Revision: None required.

Biomechanical and Physiological Mechanisms

Comment: Does the explanation of biomechanical and physiological mechanisms, such as buoyancy and hydrostatic pressure in aquatic training, provide sufficient insight into the observed outcomes?

Response: The discussion of biomechanical mechanisms, including buoyancy and hydrostatic pressure in aquatic training, has been expanded to offer deeper insight into the observed outcomes.

Revision: These points have been addressed in the discussion section.

Round 2

Reviewer 1 Report

Comments and Suggestions for Authors

I would like to thank the authors for revision of the manuscript based on reviewers suggestions. The terminology is harmonized, significant details added to methodology, information structure for the evaluation methods improved, results are are explained and conclusions reflects the findings.  

Author Response

Dear,
Thank you for your comments. They helped us improve our work.